# Effects of blast exposure on anxiety and symptoms of post-traumatic stress disorder (PTSD) among displaced Ukrainian populations

**Ken Brackstone** [1]*, **Michael G. Head** [1], **Brienna Perelli-Harris** [2]

**1** Faculty of Medicine, Clinical Informatics Research Unit, University of Southampton, Southampton, United Kingdom, **2** Department of Social Statistics and Demography, University of Southampton, Southampton, United Kingdom

* K.Brackstone@soton.ac.uk

## Abstract

Generalized anxiety and symptoms of post-traumatic stress disorder (PTSD) are common among individuals forcibly displaced during war and conflict. Blast exposure may be one important contributor of such symptoms. The aims of this study were to provide data on blast-related experiences of internally displaced persons (IDPs) and refugees following Russia's invasion of Ukraine, and to assess the influence of blast exposure on generalized anxiety, and PTSD flashbacks and nightmares. An online health needs survey was distributed to Ukrainian IDPs and refugees between April and July 2022 using Facebook Ads Manager. Participants reported whether they experienced blast exposure since the beginning of the invasion, and whether they took medication for a mental health condition before the war started. Finally, they completed measures of generalized anxiety (GAD-2), and PTSD flashbacks and nightmares. Analyses included 3253 IDPs and 5073 refugees ($N$ = 8326). Results revealed that 67.6% of total participants– 79.9% of IDPs and 61.7% of refugees–reported blast exposure since Russia's invasion. Further, 69.1% (95% CI: 68.05, 70.15) of total participants met the cut-off for generalized anxiety in which further diagnostic evaluation was warranted. Compared to refugees, IDPs reported higher generalized anxiety and greater frequency of PTSD symptoms, specifically flashbacks and nightmares. Further analyses revealed that the impact of blast exposure on flashback frequency was stronger among IDPs compared to refugees (β = 0.51; $t$(8322) = 11.88, $p$ < .0001, 95% CI: 0.43, 0.60) and among participants with pre-existing mental health conditions compared to those without (β = 0.18; $t$(8157) = 2.50, $p$ = .013, 95% CI: 0.04, 0.33). Mental health and psychosocial support must be prioritised within humanitarian relief for both IDPs and refugees and especially among people with underlying mental health conditions.

**Data Availability Statement:** Data are available on the OSF repository: https://osf.io/zm48b/.

**Funding:** This research was funded by small grants kindly provided from Public Policy@Southampton

(MH, KB, BPH), the Clinical Informatics Research Unit, University of Southampton (KB, MH), and the ESRC Centre for Population Change (BPH). The funders had no role in study design, data collection and analysis, decision to publish, or preparation of this manuscript.

**Competing interests:** The authors have declared that no competing interests exist.

## Introduction

Anxiety and post-traumatic stress disorder (PTSD) are common among war-affected communities, especially among those forcibly displaced during war and conflict [1,2]. Blast exposure may be one important contributor of such symptoms. In June 2022, 4 months after Russia began its full-scale military invasion of Ukraine on February 24 2022, reports showed that Russian military forces had delivered up to 60,000 artillery shells and rockets to Ukrainian territory every day [3]. Russia's invasion of Ukraine has created a humanitarian crisis with the fastest and largest numbers of forcibly displaced people since World War II [4]. At least 13 million people have fled their homes since the invasion began. Approximately 5 million people remain currently displaced within Ukraine itself (defined as internally displaced persons [IDPs]), while more than 8 million people have advanced for neighbouring countries in Europe and around the world (refugees) [5]. Both IDPs and refugees almost always lose housing, economic resources, and locally-embedded social connections in the event of forced displacement, all of which often lead to worsened mental health outcomes [6,7].

Blast exposure–defined as close proximity to a blast event–is a prevalent experience for many people during war, and has been linked to generalized anxiety disorder (GAD), depression, and PTSD symptoms [8,9]. It was first recognized that blast exposure could be associated with psychological and neurological symptoms reminiscent of PTSD during World War I [10]. There has been ongoing interest in the impact of blast exposure due to the conflicts in Iraq and Afghanistan. For example, 15–23% of veterans experience a form of traumatic brain injury (TBI) commonly resulting from close exposure to improvised explosive devices [11]. Even mild levels of blast exposure–defined as lower proximity to the explosion, lower duration of exposure, or milder blast forces–can cause behavioural, biochemical, and psychological effects on the nervous system, including PTSD-related behavioural traits [12]. Flashbacks, for example, are a disruptive and unpredictable symptom of PTSD that involves vividly re-experiencing traumatic events [13]. The International Classification of Diseases (ICD-11) define flashbacks on a continuum between intrusive episodes where individuals become entirely immersed in re-experiencing a trauma event and losing contact with surroundings for periods of minutes, and intrusive memories that are accompanied by a sense of reliving the event in the present, albeit fleetingly [14]. Flashback symptoms have been found in as many as one in three adult Syrian refugees living in the U.S. who experienced war-related exposure [15], whereas a high prevalence of generalized anxiety and PTSD symptoms was reported in people living in IDP camps in a war-affected Nigerian province and Sudan [16]. With high numbers of Ukrainians expected to have been exposed to blast-related violence in some capacity over the past year, it is reasonable to expect that these experiences may contribute to mental health outcomes of forcibly displaced persons now living in and outside of Ukraine.

There was a high prevalence of poor mental health amongst Ukrainians before the war due to Russia's first invasion of Ukraine in 2014, with a high prevalence of PTSD and anxiety [17]. Data collected a few weeks after Russia's invasion in 2022 found that over 50% of Ukrainians met criteria for anxiety [18], with PTSD elevated by forced displacement [19]. In fact, a systematic review recently found a high prevalence of anxiety (38.6%) and post-traumatic stress (25.7%) among civilians in war and conflicted-afflicted areas [20]. Women, in particular, are highly vulnerable to mental disorders such as PTSD, depression, and GAD during war [21,22]. In the present context, more than 50% of Ukrainian IDPs and over 90% of Ukrainian refugees are women and children, since men aged 18–60 are banned from leaving the country [23]. One possible risk factor is displacement type. Ukrainian refugees may have had more resources to flee the country, may have left earlier in the conflict, and are generally more

satisfied with housing in their new communities [24]. However, IDPs, despite being closer to family with fewer migrant and language issues, are continuously under threat and experiencing shelling, even in areas where they may have resettled. Individuals with pre-existing mental health outcomes are also more vulnerable to PTSD symptoms during war. A study found that refugees in Germany with pre-existing PTSD reported higher PTSD, generalized anxiety, and depressive symptoms immediately after experiencing another traumatic or stressful life event compared to refugees without pre-existing PTSD [25].

Other predictors of poor mental health outcomes may include forcibly displaced people with caregiving responsibilities for children or vulnerable adults, and this may be heightened if one or more of them require specialised care [26]. Older displaced people are also more susceptible to poor mental health outcomes [27]. In particular, older displaced people with health issues or disabilities may experience difficulty reaching health centres if they are dispersed in remote locations. Unfortunately, health programming of humanitarian groups in response to IDPs and refugees do not always prioritise interventions urgently needed by older people, such as assistance to disease care and prevention, or access to assistive devices [28]. Poor mental health may also manifest due to food insecurity [29], and the emotional toll of a chronic illness. For example, Syrian refugees with chronic pain near a conflict setting in the Middle East reported poor mental health one year after arriving in Northern Europe [30].

In this study, the first aim was to assess incidence of blast exposure among IDPs and refugees. The second aim was to understand differences in generalized anxiety and PTSD symptoms–flashbacks and nightmares–in IDPs and refugees, and to explore the extent in which blast exposure predicts these outcomes over and above other associated variables. The final aim was to assess the moderating influences of displacement type and diagnosed mental health conditions in predicting these outcomes after blast exposure. It was hypothesized that the association of blast exposure with generalized anxiety and PTSD flashbacks and nightmares would be stronger among IDPs compared to refugees, and among participants with pre-existing mental health conditions compared to those without. This is a crucial evidence gap that has not been explored in the context of Russia's invasion of Ukraine and subsequent mental health outcomes of IDPs and refugees.

## Methodology

### Study design and procedure

The study was an online cross-sectional survey entitled "Health Needs of Ukrainian Refugees and Internally Displaced Persons", which was advertised between April 11 and July 15 2022. All measures were translated from English into Ukrainian by a small team of experts fluent in both languages and familiar with the measures. Participants completed an online survey using Qualtrics XM. Dissemination was conducted primarily using Facebook Ads Manager. An advert describing the survey appeared on individuals' Facebook timelines along with the survey link. This technique allowed us to direct the advertising and survey link toward Ukrainian speakers. It was also disseminated via snowball methods, including Facebook shares.

### Study population

Participants were aged 18 and over who were currently living in any European country apart from Russia, or non-European countries. Participants were provided with information about the nature of the study and then provided informed consent.

## Sample size

Prior to data collection, a power analysis was conducted to determine the appropriate sample size. A confidence level of 95% and a margin error of 2% was assumed, and found that the necessary sample was 2401. The target sample size was 3000 to account for missing data. However, recruitment surged due to the timing of the survey, and a total of 10,214 participants partially or fully completed the survey.

## Measures

**Anxiety.** Anxiety was measured using the Generalized Anxiety Disorder Short-Scale [31]. Participants rated how often over the past 2 weeks they "felt nervous, anxious, or on edge" and were "not able to stop or control worrying"; 0 = *not at all*, 3 = *nearly every day*; $M = 3.59$; $SD = 1.76$); these factors were summed to form a single index ($r = .61$).

**PTSD symptoms.** Participants also rated how frequently over the past 2 weeks they had experienced flashbacks to a recently traumatic event (0 = *not at all;* 3 = *nearly every day*; $M = 2.38$; $SD = 1.03$), and nightmares about a recently traumatic event (0 = *not at all;* 3 = *nearly every day*; $M = 2.10$; $SD = 0.95$).

**Displacement type.** To categorise displacement type, a full list of countries were presented and participants indicated the country in which they were currently located. Participants were distinguished between those located outside (IDPs) and inside Ukraine (refugees).

**Blast exposure.** Participants indicated whether they had personally witnessed a blast or explosion since Russia started its invasion of Ukraine in February 2022 (yes, no).

**Health circumstances and pre-existing mental health.** Participants reported if they had any chronic health conditions that were ongoing or expected to continue 6 months or more (yes, no). Next, participants selected whether they currently had, or had been diagnosed with, depression or any other mental health condition (yes, no). If yes, participants were asked whether they were currently taking medication for their condition since some mental health conditions may have been brought on after the invasion. Thus, participants indicated if they were being treated for this condition with medication or not (yes, no). Those who reported taking medication for a mental health condition were coded into the *diagnosed mental health condition* group, and those who did not report taking medication were coded into the *no diagnosed mental health condition* group.

**Family and living situation.** Family and living variables included marital status (never married; cohabitation without marriage; married; separated but not divorced; divorced; widow/widower); whether participants had care responsibilities for adults over the age of 18 (yes, no); whether they had left anyone from their immediate family upon displacement (yes, no); current accommodation type (IPD or refugee housing, friends or relatives, other contacts or networks [e.g., friends of friends], locals offering accommodation, privately rented accommodation, hostel or hotel, other, or nowhere at the moment); and accommodation size (too small, just right, too big).

**Access to resources.** Resource variables included food access as rated on a 5-point Likert scale (1 = *poor*, 5 = *excellent*; $M = 3.82$; $SD = 0.82$); access to welfare or housing payments (yes, no); and healthcare access, which contained 4 categories coded into no access (e.g., "I currently do not have access to medical facilities and I don't know where they are [in my community]") and access (e.g., "I know where medical facilities are [in my community] and currently have access to them").

**Demographic variables.** Finally, participants indicated their age (coded into categories: 18–24, 25–34, 35–44, 45–54, 55–64, 65+), gender, language spoken at home (Ukrainian, Russian, other), Oblast in Ukraine they originated from (coded into categories: north, east, south,

west, central, Kiev), education (high [university degree or higher] and lower); and previous settlement type (rural, urban).

## Quality controls

All survey questions were translated and back-translated by four Ukrainian native speakers, and then uploaded to Qualtrics XM. The full survey was then piloted by a small number of Ukrainian speakers based in Ukraine and the UK ($n = 5$). They provided feedback on the Ukrainian language, which included phrasing of the questions and response options. These pretest responses were not included in the final analysis. All necessary adjustments to the survey, including skip logic, were made before data collection commenced.

## Ethical approval

The survey received ethical approval from University of Southampton Ethics Committee (Institutional Review Board ID: 71890). All participants provided written informed consent. All study information was written and provided on the first page of the online questionnaire, and participants indicated consent electronically by selecting the agreement box and proceeding to the survey. Individuals who did not participate received the same survey information as those who did. Participants could opt out of the study at any time during survey completion. This research was performed in accordance with the principles of the Declaration of Helsinki [32].

## Statistical analyses

Descriptive statistics and chi-square ($x^2$) analyses were first conducted to compare demographic characteristics of IDPs and refugees, and then to compare incidence of blast exposure and other categorical variables. Next, one-way ANOVAs were computed to compare generalized anxiety, flashbacks, and nightmare frequency scores between IDPs and refugees. Then, multiple linear regression models were conducted to assess for associations between independent variables–generalized anxiety, flashbacks, and nightmare frequency–among IDPs and refugees. Variable selection was performed using the backward method. Each model consisted of all factors (independent variables) with P values equal to, or smaller than, 0.2 in univariate analyses conducted beforehand. All assumptions, including tests for confounders, normality, and multicollinearity (VIFs), were tested for during model-building. Thus, any analyses that we report passed assumption checks. Six multiple linear regression models were administered for each outcome. Models included blast exposure and pre-existing mental health and demographics (Model 1), family/network (Model 2), living circumstances (Model 3), and access to resources (Model 4). Finally, a BE × Displacement Type two-way interaction (Model 5a) and a BE × Mental Health two-way interaction (Model 5b) were added to the models to assess the moderating influences of displacement type and diagnosed mental health conditions in predicting generalized anxiety and PTSD symptoms. These were analysed separately. Tests of simple slopes were subsequently administered for any significant two-way interactions using the PROCESS macro. All models were described in terms of coefficients of determination ($R^2$), which are indicative of how well the fitted models predicted the key outcome variable. Beta coefficients and p-values describing the independent variables' individual contributions to each dependent variables were recorded. All analyses were conducted using IBM SPSS Statistics Version 28.

## Results

### Demographic characteristics

Thirty-seven participants were removed for being under 18 years, and 1,851 participants were removed for not answering all measures. This left a sample of 8,326 participants who completed the survey in full ($M$ = 42.76; $SD$ = 10.68; *Age Range* = 18–93), including 3253 IDPs and 5073 refugees. Participants were mostly women (89.5%), aged 35 or over (77.6%), and of Ukrainian (95.2%) and/or Russian (20.4%) descent. Most participants completed higher education (73%), and 45.4% and 52.3% reported that their main language spoken at home was Ukrainian or Russian, respectively. 39.1% of participants were internally displaced within Ukraine and 60.9% were refugees located outside Ukraine. Among IDPs, most participants were located in eastern Ukraine (40.4%), Kyiv (29.9%), or southern Ukraine (14.4%). Among refugees, fifty-seven countries were specified among participants located outside Ukraine, with most participants located in countries within Europe (98.8%), including Poland (33.8%), Germany (17.6%), or Czech Republic (7.4%). See Table 1 for means and standard deviations of all categorical outcome measures.

### Blast exposure and mental health

IDPs (79.9%) were more likely to have experienced blast exposure compared to refugees (61.7%). IDPs were more likely to report a mental health condition (30.0%) compared to refugees (25.3%), and also to report taking medication for their condition (13.4% vs. 11.0%; Table 2). By mental health outcomes, 69.1% of total participants surpassed the GAD-2 cut-off of >3, whereby further diagnostic evaluation of generalized anxiety was warranted (95% CI: 68.05, 70.15). IDPs reported greater generalized anxiety, $F(1, 8325)$ = 71.00, $p < .0001$, $\eta^2$ = .01, and higher frequency of flashbacks, $F(1, 8325)$ = 98.17, $p < .0001$, $\eta^2$ = .01, and nightmares $F(1, 8325)$ = 53.45, $p < .0001$, $\eta^2$ = .01, compared to refugees (Table 3).

### Associations with flashback frequency

Blast exposure was strongly associated with flashback frequency, $\beta$ = 0.38; $t(8157)$ = 15.93, $p < .001$, 95% CI: 0.33, 0.43, which held after controlling for all variables, including pre-existing mental health conditions, $\beta$ = 0.30; $t(8157)$ = 8.82, $p < .001$, 95% CI: 0.23, 0.36 (Table 4; Model 4). A significant Blast Exposure × Displacement Type interaction indicated that displacement type moderated the association of blast exposure with flashback frequency, $\beta$ = -0.11; $t(8157)$ = -2.22, $p = .026$, 95% CI: -0.21, -0.01 (Fig 1; Model 5a). Tests of simple slopes revealed a significant positive association of blast exposure on flashback frequency among refugees, $\beta$ = 0.42; $t(8322)$ = 14.61, $p < .0001$, 95% CI: 0.36, 0.47, but a stronger significant positive association of blast exposure on flashback frequency among IDPs, $\beta$ = 0.51; $t(8322)$ = 11.88, $p < .0001$, 95% CI: 0.43, 0.60.

Further, a significant Blast Exposure × Mental Health interaction indicated that the pre-existing mental health conditions moderated the association of blast exposure on flashback frequency, $\beta$ = 0.18; $t(8157)$ = 2.50, $p = .013$, 95% CI: 0.04, 0.33 (Fig 2; Model 5b). Tests of simple slopes revealed a significant positive association of blast exposure on flashback frequency among participants with no pre-existing mental health conditions, $\beta$ = 0.44; $t(8322)$ = 17.94, $p < .0001$, 95% CI: 0.39, 0.49, but a stronger significant positive association of blast exposure on flashback frequency among participants with pre-existing mental health conditions, $\beta$ = 0.65; $t(8322)$ = 9.14, $p < .0001$, 95% CI: 0.51, 0.79. See Table 4 for OLS regression models and model fit coefficients.

**Table 1. Demographics of internally displaced people within Ukraine and Refugees.**

| Factors | Total—% (n) (*n* = 8326) | IDP—% (n) (*n* = 3253) | Refugee -% (n) (*n* = 5073) | $\chi^2$ (*p* value) |
|---|---|---|---|---|
| **Gender** | | | | 163.48 ($p < 0.001$) |
| Male | 10.5 (872) | 15.8 (515) | 7.0 (357) | |
| Female | 89.5 (7454) | 84.2 (2738) | 93.0 (4716) | |
| **Age** | | | | 46.26 ($p < 0.001$) |
| 18–24 | 2.9 (237) | 1.9 (60) | 3.5 (117) | |
| 25–34 | 19.6 (1633) | 18.4 (599) | 20.4 (1034) | |
| 35–44 | 38.5 (3206) | 37.5 (122) | 39.1 (1986) | |
| 45–54 | 24.0 (2001) | 26.1 (850) | 22.7 (1151) | |
| 55–64 | 11.7 (977) | 13.2 (430) | 10.8 (547) | |
| 65+ | 3.3 (272) | 2.9 (94) | 3.5 (178) | |
| **Education** | | | | 78.04 ($p < 0.001$) |
| Lower | 26.9 (2241) | 32.3 (1050) | 23.5 (1191) | |
| Higher | 73.1 (6085) | 67.7 (2203) | 76.5 (3882) | |
| **Marital status** | | | | 69.12 ($p < 0.001$) |
| Never married | 10.2 (852) | 8.1 (263) | 11.6 (589) | |
| Cohabitation | 9.0 (746) | 10.1 (330) | 8.2 (416) | |
| Married | 58.5 (4873) | 62.3 (2027) | 56.1 (2846) | |
| Separated | 2.5 (210) | 2.0 (64) | 2.9 (146) | |
| Divorced | 15.3 (1269) | 13.0 (424) | 16.6 (845) | |
| Widow/widower | 4.5 (376) | 4.5 (145) | 4.6 (231) | |
| **Main language at home** | | | | 32.37 ($p < 0.001$) |
| Ukrainian | 45.5 (3783) | 43.0 (1397) | 47.2 (2386) | |
| Russian | 52.3 (4346) | 54.3 (1763) | 51.0 (2583) | |
| Other | 2.2 (267) | 2.7 (88) | 1.8 (179) | |
| **Origin** | | | | 1036.10 ($p < 0.001$) |
| North | 6.7 (557) | 4.9 (158) | 7.9 (339) | |
| East | 40.4 (3366) | 58.9 (1916) | 28.6 (1450) | |
| South | 14.4 (1200) | 16.2 (528) | 13.2 (672) | |
| West | 4.6 (387) | 0.7 (22) | 7.2 (365) | |
| Central | 4.0 (329) | 0.8 (25) | 6.0 (304) | |
| Kyiv | 29.9 (2487) | 18.6 (604) | 37.1 (1883) | |
| **Settlement displaced from** | | | | 117.58 ($p < 0.001$) |
| City/town | 87.6 (7289) | 82.7 (2690) | 90.7 (4599) | |
| Village/rural | 12.4 (1031) | 17.3 (562) | 9.3 (469) | |
| **Completion month** | | | | 491.73 ($p < 0.001$) |
| April | 37.1 (3089) | 24.5 (796) | 45.2 (2293) | |
| May | 39.7 (3307) | 41.9 (1363) | 38.3 (1944) | |
| June | 17.0 (1414) | 24.0 (782) | 12.5 (632) | |
| July | 6.2 (516) | 9.6 (312) | 4.0 (204) | |

## Associations with generalized anxiety

Analyses revealed that blast exposure was associated with generalized anxiety, β = 0.26; *t*(8157) = 6.83, *p* < .01, 95% CI: 0.18, 0.34 (Table 5; Model 4). This association held after controlling for other variables, including pre-existing mental health conditions, β = 0.74; *t*(8157) = 12.62, *p* < .001, 95% CI: 0.63, 0.86, and displacement type, β = 0.22; *t*(8157) = 4.94, *p* < .001, 95% CI:

**Table 2. Outcome percentages of categorical outcomes comparing internally displaced people and Refugees.**

| Variables | Total—% (n) (n = 8326) | IDP—% (n) (n = 3253) | Refugee—% (n) (n = 5073) | $\chi^2$ (p value) |
|---|---|---|---|---|
| **Blast exposure** | | | | 323.62 (p < 0.001) |
| No | 32.4 (2700) | 20.9 (680) | 39.8 (2020) | |
| Yes | 67.6 (5626) | 79.1 (2573) | 60.2 (3053) | |
| **Pre-existing mental health** | | | | 10.26 (p < 0.001) |
| No | 88.0 (7331) | 86.6 (2818) | 89.0 (4513) | |
| Yes | 12.0 (995) | 13.4 (435) | 11.0 (560) | |
| **Welfare payments** | | | | 10.26 (p < 0.001) |
| No | 53.9 (4488) | 60.5 (1969) | 49.7 (2519) | |
| Yes | 46.1 (3838) | 39.5 (1284) | 50.3 (2554) | |
| **Care for over 18s** | | | | 165.43 (p < 0.001) |
| No | 80.0 (6660) | 72.9 (2373) | 84.5 (4287) | |
| Yes | 20.0 (1666) | 27.1 (880) | 15.5 (786) | |
| **Left anyone from family?** | | | | 12.11 (p < 0.001) |
| No | 7.7 (644) | 9.0 (293) | 6.9 (351) | |
| Yes | 92.3 (7682) | 91.0 (2960) | 93.1 (4722) | |
| **Chronic disease** | | | | 9.46 (p < 0.001) |
| No | 30.0 (2500) | 28.1 (914) | 31.3 (1586) | |
| Yes | 70.0 (5826) | 71.9 (2339) | 68.7 (3487) | |
| **Access to healthcare** | | | | 240.42 (p < 0.001) |
| No | 57.3 (4772) | 46.8 (1523) | 64.0 (3249) | |
| Yes | 42.7 (3554) | 53.2 (1730) | 36.0 (1824) | |
| **Where are you staying** | | | | 405.99 (p < 0.001) |
| IDP or refugee housing | 9.9 (822) | 5.8 (189) | 12.5 (633) | |
| Friends or relatives | 26.9 (2239) | 34.0 (1106) | 22.4 (1133) | |
| Friends of friends | 7.6 (631) | 7.5 (245) | 7.6 (386) | |
| Local(s) | 21.1 (1759) | 13.5 (439) | 26.0 (1320) | |
| Privately renting | 25.1 (2086) | 30.6 (994) | 21.5 (1092) | |
| Hotel or hostel | 4.2 (352) | 4.0 (129) | 4.4 (223) | |
| Other | 5.2 (437) | 4.6 (151) | 5.6 (286) | |
| **Accommodation size** | | | | 23.97 (p < 0.001) |
| Too small | 29.1 (2394) | 32.2 (1034) | 27.1 (1360) | |
| Just right/too large | 70.9 (5830) | 67.8 (2180) | 72.9 (3650) | |

0.13, 0.30. However, the subsequent Blast Exposure × Displacement Type and Blast Exposure × Mental Health interactions were nonsignificant at the 0.05 level.

## Associations with nightmares

Analyses revealed that blast exposure was associated with nightmares, β = 0.21; $t(8157) = 9.24$, $p < .001$, 95% CI: 0.17, 0.26 (Table 6; Model 4). This association held even after controlling for

**Table 3. Means and standard deviations of continuous outcomes comparing internally displaced people and Refugees.**

| Variables | Total (M ± SD) (n = 8326) | IDP (M ± SD) (n = 3253) | Refugee (M ± SD) (n = 5073) | p value |
|---|---|---|---|---|
| Anxiety (GAD-2) | 3.59 ± 1.76 | 3.80 ± 1.72 | 3.46 ± 1.78 | p < 0.001 |
| Flashback frequency | 1.38 ± 1.03 | 1.51 ± 1.03 | 1.29 ± 1.02 | p < 0.001 |
| Nightmare frequency | 1.10 ± 0.95 | 1.20 ± 0.97 | 1.04 ± 0.93 | p < 0.001 |

**Table 4. OLS regression models showing unstandardized associations between flashback frequency and independent variables.**

| | Model 1 | Model 2 | Model 3 | Model 4 | Model 5a | Model 5b |
|---|---|---|---|---|---|---|
| Blast exposure | 0.41*** | 0.41*** | 0.41*** | 0.38*** | 0.46*** | 0.36*** |
| Pre-existing mental health | 0.36*** | 0.35*** | 0.31*** | 0.30*** | 0.30*** | 0.16*** |
| **Demographics** | | | | | | |
| IDPs (ref. = refugees) | 0.12*** | 0.11*** | 0.10*** | 0.07** | 0.01 | 0.07** |
| Female (ref. = male) | 0.44*** | 0.42*** | 0.42*** | 0.41*** | 0.41*** | 0.40*** |
| Age group (ref. = 35–44) | | | | | | |
| 18–24 | 0.05 | 0.11 | 0.14* | 0.19** | 0.19** | 0.19** |
| 25–34 | -0.05 | -0.03 | -0.02 | -0.00 | -0.00 | -0.00 |
| 45–54 | 0.11*** | 0.10*** | 0.09** | 0.10*** | 0.10*** | 0.10*** |
| 55–64 | 0.21*** | 0.18*** | 0.17*** | 0.18*** | 0.18*** | 0.18*** |
| 65+ | 0.29*** | 0.27*** | 0.25*** | 0.26*** | 0.26*** | 0.26*** |
| Language at home (ref. = Ukrainian) | | | | | | |
| Russian | -0.08*** | -0.08*** | -0.08*** | -0.06** | -0.06** | -0.06** |
| Other | 0.02 | 0.03 | 0.03 | 0.02 | 0.02 | 0.03 |
| Education (ref. = high) | 0.12*** | 0.12*** | 0.12*** | 0.08*** | 0.08*** | 0.08*** |
| Origin (ref. = Kiev) | | | | | | |
| North | 0.16*** | 0.16*** | 0.16*** | 0.13** | 0.13** | 0.13** |
| East | 0.11*** | 0.11*** | 0.10** | 0.07* | 0.07* | 0.07* |
| South | 0.18*** | 0.19*** | 0.18*** | 0.16*** | 0.16*** | 0.16*** |
| West | -0.02 | -0.02 | -0.03 | -0.05 | -0.06 | -0.05 |
| Central | 0.05 | 0.05 | 0.05 | 0.02 | 0.01 | 0.02 |
| Survey completion month by participants (ref. = April) | | | | | | |
| May | -0.03 | -0.03 | -0.03 | -0.04 | -0.04 | -0.04 |
| June | 0.03 | 0.03 | 0.02 | 0.01 | 0.01 | 0.01 |
| July | -0.05 | -0.05 | -0.06 | -0.05 | -0.05 | -0.05 |
| Rural origin type (ref. = urban) | 0.12*** | 0.13*** | 0.13*** | 0.12*** | 0.12*** | 0.11*** |
| **Family/Network** | | | | | | |
| Marital status (ref. = married/cohab.) | | | | | | |
| Unmarried | | -0.08* | -0.09** | -0.10** | -0.10** | -0.10** |
| Separated/divorced/widowed | | 0.06* | 0.06* | 0.04 | 0.04 | 0.04 |
| Care responsibilities for over 18s | | 0.12*** | 0.11*** | 0.10*** | 0.10*** | 0.10*** |
| **Living circumstances** | | | | | | |
| Chronic disease | | | 0.11*** | 0.10*** | 0.10*** | 0.10*** |
| Accommodation size = small (ref. = just right or too big) | | | 0.10*** | 0.05* | 0.05* | 0.05* |
| **Access to resources** | | | | | | |
| Food | | | | -0.17*** | -0.17*** | -0.17*** |
| Healthcare access | | | | -0.09*** | -0.10*** | -0.09*** |
| **Interactions** | | | | | | |
| Blast exposure × Displacement type | | | | | -0.11* | |
| Blast exposure × Mental health | | | | | | 0.18* |
| Constant | 1.49*** | 1.49*** | 1.40*** | 2.16*** | 2.18*** | 2.17*** |
| $R^2$ | 0.101 | 0.104 | 0.109 | 0.127 | 0.128 | 0.128 |
| F | 43.85*** | 39.70*** | 38.29*** | 42.58*** | 41.30*** | 41.35*** |

*: $p < 0.05$.

**: $p < 0.01$.

***: $p < 0.001$.

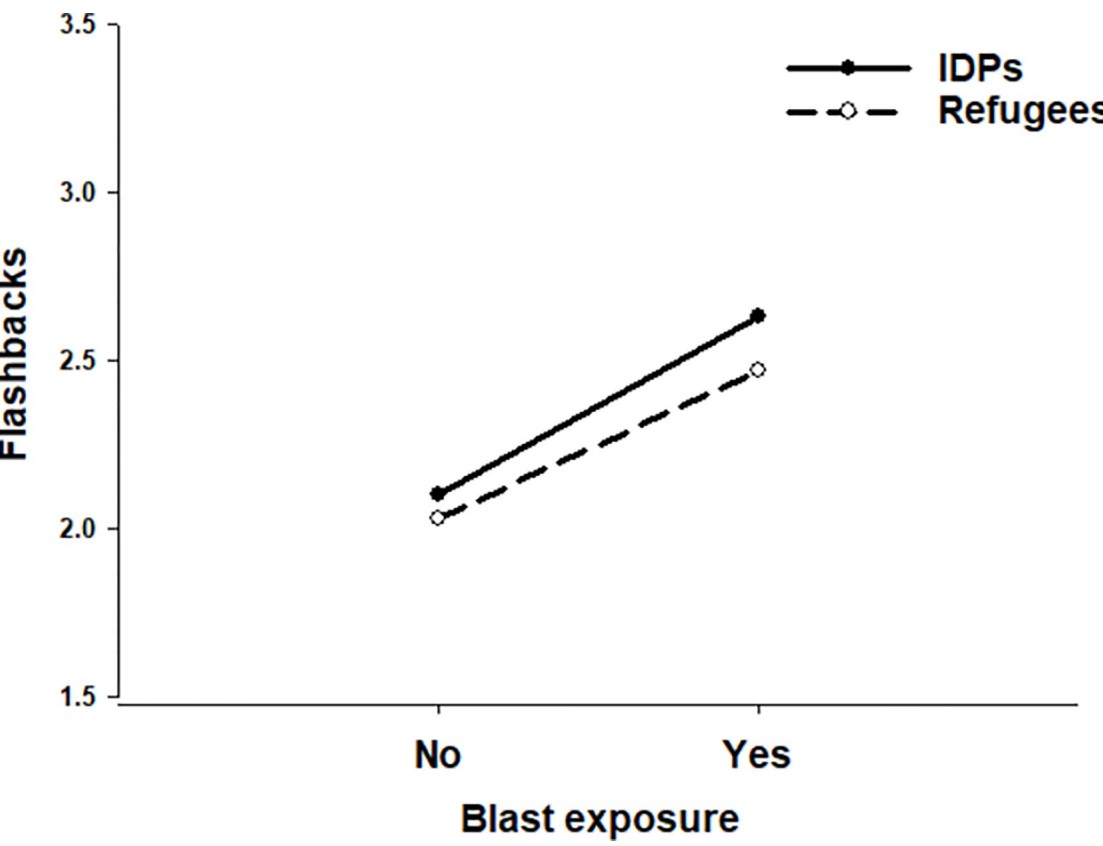

**Fig 1. Level of flashback frequency as a function of blast exposure and displacement type.**

other variables, including pre-existing mental health conditions, $\beta = 0.31$; $t(8157) = 9.75$, $p <$ .001, 95% CI: 0.25, 0.38. However, the subsequent Blast Exposure × Displacement Type and Blast Exposure × Mental Health interactions were nonsignificant at the 0.05 level.

## Discussion

These findings provide strong evidence demonstrating the extent of blast exposure on the mental health of Ukrainian citizens. Almost 70% of survey participants–approximately 80% of IDPs and over 60% of refugees–reported blast exposure between April and July 2022 since Russia's invasion began. These percentages are similar to a recent report exploring adults living in Ukraine, which found that 67.3% witnessed bombing or artillery fire during 6 months after Russia's invasion [33]. Blast exposure was strongly associated with generalized anxiety, and flashback frequency and nightmares–two symptoms commonly associated with PTSD.

Blast-exposed IDPs also reported a greater frequency of flashback symptoms compared to refugees. This is likely because IDPs are often exposed to new, or reminded of previous, blast experiences by simply living in the country. In fact, the mental health situation of IDPs appears to be much poorer compared to refugees living outside of Ukraine. These discrepancies may occur for a number of reasons. For example, compared to IDPs, refugees are more likely to live with hosts or family members, report higher satisfaction with their living conditions, and have greater access to resources and welfare support [24]. Since they are located away from the conflict, they can successfully avoid the continuous trigger of air raid sirens and the threat of

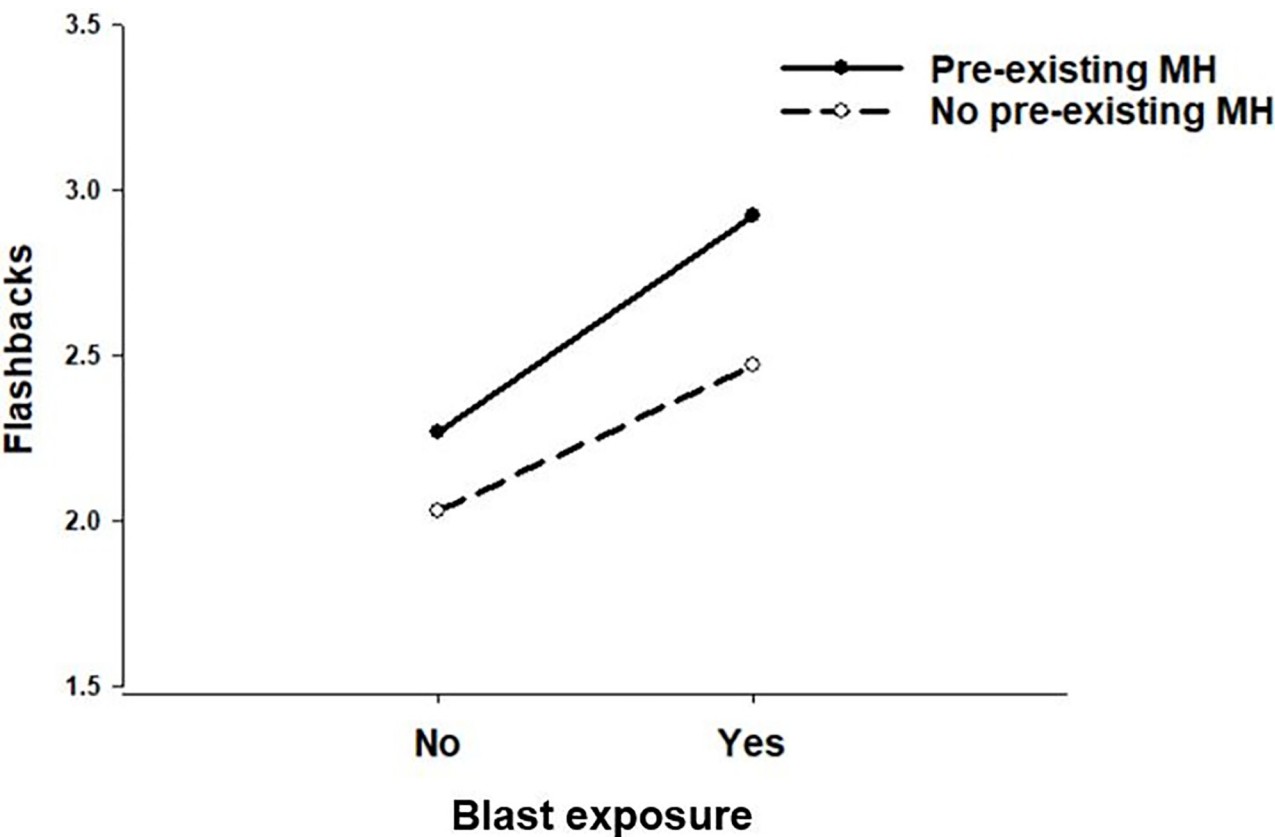

**Fig 2. Level of flashback frequency as a function of blast exposure and pre-existing mental health conditions.**

incoming shelling, which IDPs in Ukraine continue to experience on a daily basis. Despite this, it is noteworthy that blast-exposed refugees still reported a strong increase in flashback symptoms. Thus, refugees may be exposed to flashback triggers in other notable ways.

Flashback frequency was also strongest among blast-exposed people who were taking medicine for pre-existing mental health conditions. This finding fills a crucial knowledge gap demonstrating that people already diagnosed with mental health conditions may be at an increased risk of experiencing symptoms of PTSD after blast exposure. Several studies have shown that individuals with underlying mental health conditions are more vulnerable to PTSD symptoms after subsequently experiencing a stressful or traumatic event [25]. Although we did not specifically measure PTSD symptoms that they may have developed prior to Russia's invasion, some participants may have already experienced trauma after having been displaced during the 2014 separatist conflict in the eastern regions of Ukraine [34]. Around 1.5 million people were forced to leave their homes as a result of this conflict, and many settled close to the border of the separatist regions, which were and are currently war zones across 2022–23 [35]. Studies have shown that between 21% of the original Ukrainian IDP population met criteria for PTSD or Complex PTSD (CPTSD) [36]. If they fled for the second time in their lives, it is likely that their PTSD symptoms most likely would have become even more acute due to blast exposure.

Overall, approximately 69% of participants met the cut-off for generalized anxiety in which further diagnostic evaluation is warranted. This is substantially higher than previous findings shortly after the invasion showing a prevalence of around 50% [18]. Further, IDPs reported significantly higher generalized anxiety scores than refugees. However, it is worth noting that,

**Table 5. OLS regression models showing unstandardized associations between generalized anxiety and independent variables.**

| | Model 1 | Model 2 | Model 3 | Model 4 | Model 5a | Model 5b |
|---|---|---|---|---|---|---|
| Blast exposure | 0.33*** | 0.32*** | 0.31*** | 0.26*** | 0.22** | 0.26** |
| Pre-existing anxiety or depression | 0.83*** | 0.81*** | 0.77*** | 0.74*** | 0.74*** | 0.71*** |
| **Demographics** | | | | | | |
| IDPs (ref. = refugees) | 0.33*** | 0.30*** | 0.30*** | 0.22*** | 0.27*** | 0.22*** |
| Female (ref. = male) | 0.94*** | 0.94*** | 0.94*** | 0.92*** | 0.92*** | 0.92*** |
| Age group (ref. = 35–44) | | | | | | |
| 18–24 | 0.10 | 0.10 | 0.15 | 0.24* | 0.24* | 0.24* |
| 25–34 | -0.04 | -0.04 | -0.02 | 0.01 | 0.01 | 0.01 |
| 45–54 | 0.00 | 0.00 | -0.01 | 0.01 | 0.01 | 0.00 |
| 55–64 | 0.01 | 0.03 | 0.04 | 0.04 | 0.04 | 0.04 |
| 65+ | 0.03 | 0.06 | 0.07 | 0.08 | 0.08 | 0.08 |
| Language at home (ref. = Ukrainian) | | | | | | |
| Russian | -0.07 | -0.07 | -0.07 | -0.03 | -0.03 | -0.03 |
| Other | 0.05 | 0.06 | 0.06 | 0.05 | 0.05 | 0.05 |
| Education (ref. = high) | 0.02 | 0.03 | 0.03 | -0.04 | -0.04 | -0.04 |
| Origin (ref. = Kiev) | | | | | | |
| North | -0.12 | -0.11 | -0.11 | -0.16* | -0.16* | -0.16* |
| East | 0.12* | 0.13** | 0.11* | 0.07 | 0.07 | 0.07 |
| South | 0.21*** | 0.22*** | 0.22*** | 0.17** | 0.17** | 0.17** |
| West | 0.15 | 0.16 | 0.16 | 0.12 | 0.12 | 0.11 |
| Central | -0.01 | -0.00 | -0.01 | -0.06 | -0.05 | -0.06 |
| Survey completion month by participants (ref. = April) | | | | | | |
| May | -0.22*** | -0.22*** | -0.22*** | -0.22*** | -0.22*** | -0.22*** |
| June | -0.12* | -0.11 | -0.12* | -0.12* | -0.12* | -0.12* |
| July | -0.23** | -0.23** | -0.24** | -0.20* | -0.20* | -0.20* |
| **Family/Network** | | | | | | |
| Marital status (ref. = married/cohab.) | | | | | | |
| Unmarried | | -0.01 | -0.03 | -0.05 | -0.05 | -0.05 |
| Separated/divorced/widowed | | -0.08 | -0.08 | -0.12** | -0.12** | -0.12** |
| Care responsibilities for over 18s | | 0.24*** | 0.22*** | 0.20*** | 0.20*** | 0.20*** |
| Left anyone from immediate family | | 0.12 | 0.12 | 0.10 | 0.10 | 0.10 |
| **Living circumstances** | | | | | | |
| Chronic disease | | | 0.10* | 0.08 | 0.08 | 0.08 |
| Accommodation size = small (ref. = just right or too big) | | | 0.28*** | 0.18*** | 0.18*** | 0.18*** |
| **Access to resources** | | | | | | |
| Food | | | | -0.32*** | -0.32*** | -0.32*** |
| Governmental welfare payments | | | | -0.06 | -0.06 | -0.07 |
| Healthcare access | | | | -0.19*** | -0.19*** | -0.19*** |
| **Interactions** | | | | | | |
| Blast exposure × Displacement type | | | | | 0.07 | |
| Blast exposure × Mental health | | | | | | 0.04 |
| Constant | 2.37*** | 2.24*** | 2.12*** | 3.62*** | 3.61*** | 3.63*** |
| $R^2$ | 0.073 | 0.077 | 0.083 | 0.107 | 0.108 | 0.107 |
| $F$ | 32.23*** | 28.24*** | 28.28*** | 33.81*** | 32.70*** | 32.68*** |

*: $p < 0.05$.

**: $p < 0.01$.

***: $p < 0.001$.

**Table 6. OLS regression models showing unstandardized associations between nightmares and independent variables.**

|  | Model 1 | Model 2 | Model 3 | Model 4 | Model 5a | Model 5b |
|---|---|---|---|---|---|---|
| Blast exposure | 0.24*** | 0.24*** | 0.23*** | 0.21*** | 0.21*** | 0.21*** |
| Pre-existing mental health | 0.37*** | 0.36*** | 0.33*** | 0.31*** | 0.31*** | 0.29*** |
| **Demographics** |  |  |  |  |  |  |
| IDPs (ref. = refugees) | 0.12*** | 0.11*** | 0.11*** | 0.07** | 0.07 | 0.07** |
| Female (ref. = male) | 0.26*** | 0.26*** | 0.26*** | 0.25*** | 0.25*** | 0.25*** |
| Age group (ref. = 35–44) |  |  |  |  |  |  |
| 18–24 | -0.11 | -0.10 | -0.08 | -0.03 | -0.03 | -0.03 |
| 25–34 | 0.03 | 0.03 | 0.04 | 0.06* | 0.06* | 0.06* |
| 45–54 | -0.08** | -0.08** | -0.10*** | -0.09*** | -0.09*** | -0.09*** |
| 55–64 | -0.21*** | -0.20*** | -0.21*** | -0.21*** | -0.21*** | -0.21*** |
| 65+ | -0.16** | -0.16** | -0.17** | -0.17** | -0.17** | -0.17** |
| Language at home (ref. = Ukrainian) |  |  |  |  |  |  |
| Russian | -0.10*** | -0.10*** | -0.09*** | -0.08*** | -0.08*** | -0.07*** |
| Other | -0.14 | -0.13 | -0.12 | -0.14 | -0.14 | -0.14 |
| Education (ref. = high) | 0.09*** | 0.09*** | 0.09*** | 0.06* | 0.06* | 0.06* |
| Origin (ref. = Kiev) |  |  |  |  |  |  |
| North | 0.11* | 0.11* | 0.11* | 0.09 | 0.09 | 0.08 |
| East | 0.04 | 0.04 | 0.04 | 0.02 | 0.02 | 0.02 |
| South | 0.10** | 0.10** | 0.10** | 0.08* | 0.08* | 0.08* |
| West | 0.03 | 0.03 | 0.03 | 0.01 | 0.01 | 0.01 |
| Central | -0.02 | -0.02 | -0.02 | -0.04 | -0.04 | -0.04 |
| Survey completion month by participants (ref. = April) |  |  |  |  |  |  |
| May | -0.04 | -0.03 | -0.03 | -0.04 | -0.04 | -0.04 |
| June | -0.02 | -0.02 | -0.02 | -0.02 | -0.02 | -0.02 |
| July | -0.07 | -0.07 | -0.07 | -0.05 | -0.05 | -0.05 |
| Rural origin type (ref. = urban) | 0.04 | 0.05 | 0.05 | 0.03 | 0.03 | 0.03 |
| **Family/Network** |  |  |  |  |  |  |
| Marital status (ref. = married/cohab.) |  |  |  |  |  |  |
| Unmarried |  | -0.01 | -0.01 | -0.03 | -0.03 | -0.03 |
| Separated/divorced/widowed |  | 0.00 | 0.00 | -0.02 | -0.02 | -0.02 |
| Care responsibilities for over 18s |  | 0.09*** | 0.08** | 0.07** | 0.07** | 0.07** |
| Left anyone from immediate family |  | 0.08* | 0.08* | 0.07 | 0.07 | 0.07 |
| **Living circumstances** |  |  |  |  |  |  |
| Chronic disease |  |  | 0.10*** | 0.09*** | 0.09*** | 0.09** |
| Accommodation size = small (ref. = just right or too big) |  |  | 0.08*** | 0.03 | 0.03 | 0.03 |
| **Access to resources** |  |  |  |  |  |  |
| Food |  |  |  | -0.15*** | 0.15*** | 0.15*** |
| Governmental welfare payments |  |  |  | -0.04 | -0.04 | -0.03 |
| Healthcare access |  |  |  | -0.10*** | -0.10*** | -0.10*** |
| **Interactions** |  |  |  |  |  |  |
| Blast exposure × Displacement type |  |  |  |  | -0.00 |  |
| Blast exposure × Mental health |  |  |  |  |  | -0.03 |
| Constant | 1.67*** | 1.58*** | 1.51*** | 2.24*** | 2.24*** | 2.24*** |
| $R^2$ | 0.057 | 0.059 | 0.062 | 0.082 | 0.082 | 0.082 |
| F | 23.46*** | 20.36*** | 20.00*** | 24.30*** | 23.56*** | 23.57*** |

*: $p < 0.05$.

**: $p < 0.01$.

***: $p < 0.001$.

whilst blast exposure was strongly associated with generalized anxiety and nightmares, other variables were more strongly associated with these outcomes. For example, replicating previous findings in displaced people, food insecurity was a stronger predictor of generalized anxiety, similar to other research recently conducted among Syrian refugees in Norway [29], and women participants were a stronger predictor of PTSD nightmare symptoms. Thus, blast exposure is clearly one part of the mental health picture.

## Implications

Despite a high prevalence of poor mental health amongst Ukrainians before 2022 [17], this study demonstrates that Russia's invasion of Ukraine has resulted in other adverse mental health outcomes pertaining to symptoms of PTSD. This corroborates recent work showing that 73.2% of Ukrainian civilians met the diagnostic criteria for PTSD since Russia's invasion of Ukraine began [37]. Flashbacks, in particular, are a disruptive and unpredictable symptom of PTSD that involves vividly re-experiencing traumatic events for periods of minutes [13]. They are often accompanied by a sense of reliving a past event in the present, albeit fleetingly [14]. High numbers of Ukrainians are exposed to blast-related violence, and these experiences will undoubtedly continue to contribute to adverse mental health outcomes of forcibly displaced Ukrainians as the war progresses.

Targeted and scalable interventions or psychological support groups should be implemented by community mental health services within Ukraine and refugee countries to address these symptoms and improve overall mental health and wellbeing [38]. Digital cognitive-behavioural therapy (CBT), for example, has been found to support the short- and long-term treatment of insomnia–a commonly-reported PTSD symptom [39]. Due to the high level of accessibility of digital CBT for patients worldwide, this should be one option offered to all Ukrainians who may have experienced war-related violence and blast exposure. This study also provides further evidence to underpin the need for programmes such as the WHO Special Initiative for Mental Health, which should be prioritised during Ukraine's process of resilience and recovery from this conflict [40].

## Limitations and future research

This survey has many strengths, including rapid implementation early on in the conflict, and a large sample size which provides practical insight during a challenging humanitarian situation in which real-time data is often difficult to gather. However, because participants were not selected based on a sampling framework, the survey may not be representative of the displaced and refugee Ukrainian population. Participants were required to have access to an electronic device and the internet, and vulnerable persons may not have had access at the time of data collection. Facebook use in Ukraine is biased toward middle aged women who are better educated [41]. People in the west and centre of the country were more likely to use Facebook than those in eastern regions, where the conflict has been most acute and forced more people to leave [41]. The survey was also advertised as a health survey, thus individuals with health conditions may have been more likely to participate. Nevertheless, estimates of gender and education levels are in line with other nationally representative surveys conducted on Ukrainian refugees since the invasion, including refugees in Germany [42].

Due to limitations in the number of questions participants could realistically be expected to answer, some outcome variables were not assessed in the most rigorous way. For example, flashbacks are typically a response to internal or external cues. Issuing one item about the frequency of flashbacks during the past 2 weeks may not have been the most accurate method for assessing them. Retrospective judgements can often be unreliable, and some individuals may

temporarily suppress symptoms using avoidance techniques [43]. Further, there was no measure of baseline anxiety from before Russia's invasion of Ukraine, which would have been crucial in capturing the true impact of displacement on mental health outcomes. Thus, there are no prospective measures for change, and large-scale surveys in Ukraine are rare, with data collection stalling in the 2010s. A series of follow-up surveys to assess levels of generalized anxiety and PTSD symptoms across the course of the war are currently being administered. Future studies should also ask about previous displacement to better understand the link between multiple traumatic events, flashbacks, and mental health conditions.

Finally, PTSD could have been operationalized using more sophisticated trauma questionnaires, such as the International Trauma Questionnaire (ITQ) [44]. With blast-related exposure strongly associated with mental health outcomes, additional questions relating to the context of the blast exposure would have been pertinent understand how specific contexts relate to poor mental health outcomes and psychological trauma. For example, proximity to blasts is often related to mental health symptoms [45]. Additional questions relating to the circumstances for their displacement (e.g., whether or not participants were forced to evacuate because their homes were destroyed) are also important in relation to mental health. Extra attention should be paid to these types of nuances in future research, and especially during the active phases of war [46].

## Conclusions

The Ukraine displacement crisis is arguably the highest-profile crisis of its kind globally. This study indicates that approximately 70% of forcibly displaced Ukrainian citizens experienced blast exposure and deteriorated mental health during the first few months of conflict. Despite a high prevalence of poor mental health amongst Ukrainians before the war [17], IDPs appeared to fare worse than refugees on a variety of outcomes. Blast exposure was especially influential in predicting PTSD flashback symptoms, particularly among IDP populations and individuals with pre-existing mental health conditions. There is therefore an urgent need to understand the nuances of trauma experiences, and the effect that they may have on mental health outcomes–especially among individuals forcibly displaced as a result of war and conflict.

## Supporting information

**S1 Checklist. STROBE statement—Checklist of items that should be included in reports of cross-sectional studies.**
(DOCX)

**S2 Checklist. Inclusivity in global research.**
(DOCX)

**S1 Table. OLS regression models showing standardized associations between flashback frequency and independent variables.**
(DOCX)

**S2 Table. OLS regression models showing standardized associations between generalized anxiety and independent variables.**
(DOCX)

**S3 Table. OLS regression models showing standardized associations between nightmares and independent variables.**
(DOCX)

## Acknowledgments

The authors would like to thank our friends and colleagues for help with translation and interpretation: Yuliya Hilevych, Nataliia Levchuk, Margaryta Dorosh, Ksenia Crane, and Inna Walker.

## Author Contributions

**Conceptualization:** Ken Brackstone, Michael G. Head, Brienna Perelli-Harris.

**Data curation:** Ken Brackstone, Michael G. Head, Brienna Perelli-Harris.

**Formal analysis:** Ken Brackstone, Brienna Perelli-Harris.

**Investigation:** Ken Brackstone, Michael G. Head, Brienna Perelli-Harris.

**Methodology:** Ken Brackstone, Michael G. Head, Brienna Perelli-Harris.

**Project administration:** Ken Brackstone, Michael G. Head.

**Writing – original draft:** Ken Brackstone.

**Writing – review & editing:** Ken Brackstone, Michael G. Head, Brienna Perelli-Harris.

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
