## [Decision Letter · Decision Letter 0]

22 Sep 2023

PGPH-D-23-00922

Blast Exposure and Post-Traumatic Stress Disorder (PTSD) in Ukrainian Displaced Persons: The Influences of Displacement Type and Pre-Existing Mental Health Conditions

Dear Dr. Brackstone,

Thank you for submitting your manuscript to PLOS Global Public Health. After careful consideration, we feel that it has merit but does not fully meet PLOS Global Public Health’s publication criteria as it currently stands. Therefore, we invite you to submit a revised version of the manuscript that addresses the points raised during the review process.

We look forward to receiving your revised manuscript.

Kind regards,

Humayun Kabir

Academic Editor

Comments from PLOS Editorial Office: We note that one or more reviewers has recommended that you cite specific previously published works. As always, we recommend that you please review and evaluate the requested works to determine whether they are relevant and should be cited. It is not a requirement to cite these works. We appreciate your attention to this request.

Journal Requirements:

1. Please provide additional information regarding the considerations  made for the refugees included in this study. For instance, please discuss whether participants were able to opt out of the study and whether individuals who did not participate receive the same treatment offered to participants.

2. Our staff editors have determined that your manuscript is likely within the scope of our Global Mental Health: challenges, opportunities, and the future of the field. This editorial initiative is headed by a team of Guest Editors for PLOS GPH: Rochelle Burgess (University College of London) and Dixon Chibanda (University of Zimbabwe and London School of Tropical Medicine and Hygiene). The Collection invites researchers to submit original research which engages with, or disrupts, the urgent needs across the global mental health landscape. We especially encourage submissions of studies that critically interrogate the status quo of the field and that involve inter-/trans-disciplinary approaches and those which share perspectives from underrepresented global regions and communities.

 Additional information can be found on our announcement page: https://collections.plos.org/call-for-papers/global-mental-health-opportunities-challenges/ 

If you would like your manuscript to be considered for this collection, please let us know in your cover letter and we will ensure that your paper is treated as if you were responding to this call.  Please note that being considered for the Collection does not require additional peer review beyond the journal’s standard process and will not delay the publication of your manuscript if it is accepted by PLOS GPH. If you would prefer to remove your manuscript from collection consideration, please specify this in the cover letter.

3. Please amend your detailed Financial Disclosure statement. This is published with the article. It must therefore be completed in full sentences and contain the exact wording you wish to be published.

4. Please provide separate figure files in .tif or .eps format only and remove any figures embedded in your manuscript file. Please also ensure all files are under our size limit of 10MB.

Additional Editor Comments (if provided):

Reviewers' comments:

Reviewer's Responses to Questions

**Comments to the Author**

1. Does this manuscript meet PLOS Global Public Health’s publication criteria? Is the manuscript technically sound, and do the data support the conclusions? The manuscript must describe methodologically and ethically rigorous research with conclusions that are appropriately drawn based on the data presented.

Reviewer #1: Yes

Reviewer #2: Yes

Reviewer #3: Partly

Reviewer #4: Partly

2. Has the statistical analysis been performed appropriately and rigorously?

Reviewer #1: Yes

Reviewer #2: Yes

Reviewer #3: No

Reviewer #4: No

3. Have the authors made all data underlying the findings in their manuscript fully available (please refer to the Data Availability Statement at the start of the manuscript PDF file)?

Reviewer #1: Yes

Reviewer #2: Yes

Reviewer #3: Yes

Reviewer #4: Yes

4. Is the manuscript presented in an intelligible fashion and written in standard English?

Reviewer #1: Yes

Reviewer #2: Yes

Reviewer #3: Yes

Reviewer #4: Yes

5. Review Comments to the Author

Reviewer #1: I have the following comments for the authors to address.

1) Please refer to the folllowing finding in the Introduction as part of the literature review:

Prevalence of depression, anxiety and post-traumatic stress in war- and conflict-afflicted areas: A meta-analysis. Front Psychiatry. 2022 Sep 16;13:978703. doi: 10.3389/fpsyt.2022.978703. PMID: 36186881; PMCID: PMC9524230.

2) The authors can discuss the finding of the following paper and how their findings are similar or different.

Depression, anxiety and post-traumatic stress during the 2022 Russo-Ukrainian war, a comparison between populations in Poland, Ukraine, and Taiwan. Sci Rep. 2023 Mar 3;13(1):3602. doi: 10.1038/s41598-023-28729-3. PMID: 36869035; PMCID: PMC9982762.

3) Under discussion, please recommend the use of Internet or digit psychotherapy to help Ukrainians:

Search PubMed for: This meta-analysis provides strong support for the effectiveness of dCBT-I in treating insomnia. dCBT-I has potential to revolutionise the delivery of CBT-I, improving the accessibility and availability of CBT-I content for insomnia patients worldwide.

Reviewer #2: General Comments

1. Use third person and not first or second person in sentence construction – see line 32, 33, 35, 138, 163

Specific comments

Suggest title be improved as: Effects of Blast exposure on Post-Traumatic Stress Disorder (PTSD) among displaced Ukrainian population

Abstract

•Structure it so that the following subsections are visible: Background, Aim, Methods, Result and conclusion and then merged in one as opposed to sub-sections

• Abstract is quite big

• First sentence in abstract and Introduction should address the problem investigated which is related to mental illnesses and PTSD before address the cause in this case “Blast Exposures”.

Introduction

• Aims: Line 138-147- Aims need to be shorted and straight forward and therefore need to be re-written

Methods

• Line 148 : Subtitle should be “Methodology”

• More information is needed on the Methodology.

• Methods are not flowing and please structure it as below:

o Study design

o Study setting

o Study population

o Sample size

o Selection criteria (Inclusion and exclusion)

o Tools used

o Sampling procedures

o Procedures for data collection

o Data management and quality control

o Statistical data analysis

o Ethical consideration

Results

• Results to be sub-sectioned and given clear sub-titles

• Sub-section should be “Demographic characteristics”

• Line 239 -Table 1: Demographics of Internally Displaced People within Ukraine and Refugees, Percentages should in the sub-title at the top of the cell in each column and not on every figure and % should come first and then (n) i.e. % (n)

• The % seems not to add up since they are calculated on each item. Suggest that it should be calculated based on the total i.e. Total Gender is n=8326 (100%) for both IDP and Refugees. What is the % contribution of male and female for IDP and refugees based on the total number such that when added up they give 100%. This should be done all through the table.

• Line 254 -Table 2: Outcome Percentages of Internally Displaced People within Ukraine and Refugees Categorical Outcomes (M ± SD) ----- See table 1 above. % should not be on every figure. Let it be put on the sub-title in each column

• And % should add up as commented in Table 1 above

• Line 258 - Table 3: Outcome Percentages of Internally Displaced People within Ukraine and Refugees Continuous Outcomes (M ± SD) – Are the figure reported % or something else? What are the units of those figures?

• Line 274- Table 4: OLS regression models showing associations between flashback frequency and independent variables – Table needs a key. What is ****? What does – and + values in the table mean in the regression analysis?

• Line 319- Table 5. OLS regression models showing associations between generalized anxiety and independent variables -- Table needs a key. What is ****? What does – and + values in the table mean in the regression analysis?

• Line 359 - Table 6. OLS regression models showing associations between nightmares and independent variables --- Table needs a key. What is ****? What does – and + values in the table mean in the regression analysis?

Discussion

• I suggest that Discussion be sub-sectioned according to the results.

• Use third person and not first or second person in sentence construction

• Discussion should related to findings and the tables should be quoted

• What is the role of Blasts exposure to Mental illnesses among the new cases and those with existing cases?

Conclusion

• Use third person and not first or second person in sentence construction

• Conclusion should be made on each of the key findings on each sub section

Reviewer #3: PGPH-D-23-00922

Please make the title smaller

Break the abstract in separate parts like introduction/background, methods, results, conclusion.

Please improve the justification of the study.

Add why this study is needed in the context.

I can not find any tool in the method that were used to assess the PTSD.

If the PTSD is not assessed, then how author say PTSD in the title. I think should be careful about their study planning and reporting.

Follow the reporting guidelines STOBE and cover all the sections in the STOBE.

Give description of the population.

Add a section for the questionnaire development.

Add a section for the data collection techniques.

Add a section for the bias.

Add a section for the dependent variables and independent variables.

Add the exclusion and inclusion criteria.

Report the sampling techniques.

Report the sample size calculation.

Repot the validity and reliability of the tools used in this study.

Cite the literature from where author used the tools such as measure of mental health, displacement types and so on.

Provide equitable description and the scoring of the tools.

Discussion should be more contractive.

Add sections:

• Limitations of the study

• Recommendation for the stakeholders

• What is new in this study?

• Future research

Reviewer #4: “First, chi-square (x²) analyses were used to examine associations between

217 displacement type and demographics and outcome predictors” give all rationale of the test.

Please specify when the tests were done.

All the tests should be consistent with the study objective.

How were the variables selected in the adjusted model, please meet the selection criteria?

Did the assumptions of the model fitted, report the all the assumptions of the model?

did author check for confounding, Effect modification, multi-collinearity?

As the author used linear regression, were the assumptions met, importantly normality, and multicollinearity?

The author can not fit the model (putting all variables in the model) without following the model-building criteria.

In the regression model, variable selection requires justification like the check of confounding, interaction, and association in bivariate regression along with the support of the literature.

After fitting the model, did they check the model's fitness? There are many ways to check the model fitness, and predictive accuracies, which require providing the information. Submit the figures (for example Q-Q plots) of the model accuracies as supplementary files if required.

Require reporting the post-estimation test VIF.

Research question requires outlines in the justification and the model should be built to address these questions.

6. PLOS authors have the option to publish the peer review history of their article (what does this mean?). If published, this will include your full peer review and any attached files.

**Do you want your identity to be public for this peer review?** For information about this choice, including consent withdrawal, please see our Privacy Policy.

Reviewer #1: No

Reviewer #2: **Yes: **DR. Godfrey S. Bbosa (MSc. Pharm, MSc.IPHC, M. Neurosci, PhD)

Reviewer #3: No

Reviewer #4: No

---

## [Decision Letter · Decision Letter 1]

13 Dec 2023

PGPH-D-23-00922R1

Effects of Blast Exposure on Anxiety and Symptoms of Post-Traumatic Stress Disorder (PTSD) among Displaced Ukrainian Populations

Dear Dr. Brackstone,

Thank you for submitting your manuscript to PLOS Global Public Health. After careful consideration, we feel that it has merit but does not fully meet PLOS Global Public Health’s publication criteria as it currently stands. Therefore, we invite you to submit a revised version of the manuscript that addresses the points raised during the review process.

We look forward to receiving your revised manuscript.

Kind regards,

Humayun Kabir

Academic Editor

Journal Requirements:

1. Please update your online Competing Interests statement. If you have no competing interests to declare, please state: “The authors have declared that no competing interests exist.”

Additional Editor Comments (if provided):

Reviewers' comments:

Reviewer's Responses to Questions

**Comments to the Author**

1. If the authors have adequately addressed your comments raised in a previous round of review and you feel that this manuscript is now acceptable for publication, you may indicate that here to bypass the “Comments to the Author” section, enter your conflict of interest statement in the “Confidential to Editor” section, and submit your "Accept" recommendation.

Reviewer #1: All comments have been addressed

Reviewer #2: All comments have been addressed

2. Does this manuscript meet PLOS Global Public Health’s publication criteria? Is the manuscript technically sound, and do the data support the conclusions? The manuscript must describe methodologically and ethically rigorous research with conclusions that are appropriately drawn based on the data presented.

Reviewer #1: Yes

Reviewer #2: Yes

3. Has the statistical analysis been performed appropriately and rigorously?

Reviewer #1: Yes

Reviewer #2: Yes

4. Have the authors made all data underlying the findings in their manuscript fully available (please refer to the Data Availability Statement at the start of the manuscript PDF file)?

Reviewer #1: Yes

Reviewer #2: Yes

5. Is the manuscript presented in an intelligible fashion and written in standard English?

Reviewer #1: Yes

Reviewer #2: Yes

6. Review Comments to the Author

Reviewer #1: I recommend publication.

Reviewer #2: Line 139: Use past tense

Study design and setting: Study setting is not included

Line 158: There should be “Study data collection procedures” before you go to Measures

In tables: < .001, write it as “p<0.001

Line 263: write < .001 as “p<0.001

7. PLOS authors have the option to publish the peer review history of their article (what does this mean?). If published, this will include your full peer review and any attached files.

**Do you want your identity to be public for this peer review?** For information about this choice, including consent withdrawal, please see our Privacy Policy.

Reviewer #1: No

Reviewer #2: **Yes: **Dr. Godfrey S. Bbosa

---

## [Decision Letter · Decision Letter 2]

15 Feb 2024

Effects of Blast Exposure on Anxiety and Symptoms of Post-Traumatic Stress Disorder (PTSD) among Displaced Ukrainian Populations

PGPH-D-23-00922R2

Dear Ken Brackstone,

We are pleased to inform you that your manuscript 'Effects of Blast Exposure on Anxiety and Symptoms of Post-Traumatic Stress Disorder (PTSD) among Displaced Ukrainian Populations' has been provisionally accepted for publication in PLOS Global Public Health.

Best regards,

Bibhav Acharya

Academic Editor

All concerns from reviewers have been addressed

Reviewer Comments (if any, and for reference):

Reviewer's Responses to Questions

**Comments to the Author**

1. If the authors have adequately addressed your comments raised in a previous round of review and you feel that this manuscript is now acceptable for publication, you may indicate that here to bypass the “Comments to the Author” section, enter your conflict of interest statement in the “Confidential to Editor” section, and submit your "Accept" recommendation.

Reviewer #1: All comments have been addressed

Reviewer #2: All comments have been addressed

2. Does this manuscript meet PLOS Global Public Health’s publication criteria? Is the manuscript technically sound, and do the data support the conclusions? The manuscript must describe methodologically and ethically rigorous research with conclusions that are appropriately drawn based on the data presented.

Reviewer #1: Yes

Reviewer #2: Yes

3. Has the statistical analysis been performed appropriately and rigorously?

Reviewer #1: Yes

Reviewer #2: Yes

4. Have the authors made all data underlying the findings in their manuscript fully available (please refer to the Data Availability Statement at the start of the manuscript PDF file)?

Reviewer #1: Yes

Reviewer #2: Yes

5. Is the manuscript presented in an intelligible fashion and written in standard English?

Reviewer #1: Yes

Reviewer #2: Yes

6. Review Comments to the Author

Reviewer #1: I recommend publication.

Reviewer #2: Line 138: Study design and procedure - It should be “Study design and setting”

Study procedure – Should be below Sample size sub-section – Line 152

Line 277: Table 3. Means and Standard Deviations scores of Continuous Outcomes Comparing Internally Displaced People and Refugees

Add word “scores” in subtitle and the table

7. PLOS authors have the option to publish the peer review history of their article (what does this mean?). If published, this will include your full peer review and any attached files.

**Do you want your identity to be public for this peer review?** For information about this choice, including consent withdrawal, please see our Privacy Policy.

Reviewer #1: No

Reviewer #2: No
